# Memantine Disrupts Motor Coordination through Anxiety-like Behavior in CD1 Mice

**DOI:** 10.3390/brainsci12040495

**Published:** 2022-04-13

**Authors:** Anton N. Shuvaev, Olga S. Belozor, Oleg I. Mozhei, Aleksandra G. Mileiko, Ludmila D. Mosina, Irina V. Laletina, Ilia G. Mikhailov, Yana V. Fritsler, Andrey N. Shuvaev, Anja G. Teschemacher, Sergey Kasparov

**Affiliations:** 1Research Institute of Molecular Medicine and Pathobiochemistry, Krasnoyarsk State Medical University Named after Prof. V.F. Voino-Yasenetsky, 660022 Krasnoyarsk, Russia; olsbelor@gmail.com; 2Institute of Living Systems, Immanuel Kant Baltic Federal University, 236041 Kaliningrad, Russia; vulpecula999@gmail.com (O.I.M.); sergey.kasparov@bristol.ac.uk (S.K.); 3Institute of Fundamental Biology and Biotechnology, Siberian Federal University, 660041 Krasnoyarsk, Russia; sashamileiko@mail.ru (A.G.M.); l-mosina00@mail.ru (L.D.M.); laletina.irina.katrinriel16@gmail.com (I.V.L.); medw.medwed2015@yandex.ru (I.G.M.); fri.yana@mail.ru (Y.V.F.); andrey.n.shuvaev@gmail.com (A.N.S.); 4Department of Physiology, Pharmacology, and Neuroscience, University of Bristol, Bristol BS8 1TD, UK; anja.teschemacher@bristol.ac.uk

**Keywords:** ataxia, memantine, anxiety, rotarod

## Abstract

Memantine is an FDA approved drug for the treatment of Alzheimer’s disease. It reduces neurodegeneration in the hippocampus and cerebral cortex through the inhibition of extrasynaptic NMDA receptors in patients and mouse models. Potentially, it could prevent neurodegeneration in other brain areas and caused by other diseases. We previously used memantine to prevent functional damage and to retain morphology of cerebellar neurons and Bergmann glia in an optogenetic mouse model of spinocerebellar ataxia type-1 (SCA1). However, before suggesting wider use of memantine in clinics, its side effects must be carefully evaluated. Blockers of NMDA receptors are controversial in terms of their effects on anxiety. Here, we investigated the effects of chronic application of memantine over 9 weeks to CD1 mice and examined rotarod performance and anxiety-related behaviors. Memantine-treated mice exhibited an inability to adapt to anxiety-causing conditions which strongly affected their rotarod performance. A tail suspension test revealed increased signs of behavioral despair. These data provide further insights into the potential deleterious effects of memantine which may result from the lack of adaptation to novel, stressful conditions. This effect of memantine may affect the results of tests used to assess motor performance and should be considered during clinical trials of memantine in patients.

## 1. Introduction

Glutamate is the principal neurotransmitter which mediates most excitatory synaptic activity in the central nervous system. Long-term plasticity of glutamatergic synapses is thought to underpin memory formation, a process in which NMDA receptors are particularly important. On the other hand, it is generally accepted that, in certain neurodegenerative conditions, overactivity of excitatory transmission leads to excessive influx of sodium and calcium ions into neurons, resulting in excitotoxicity, i.e., cell damage and death [1]. The most important pathway for Ca^2+^ entry and, therefore, excitotoxicity are NMDA receptors, which have been implicated in pathologies such as traumatic brain injury and Alzheimer’s disease [2]. Following this rationale, memantine, a non-competitive, voltage-dependent NMDA receptor antagonist, should prevent NMDA receptor mediated excitotoxicity and neurodegeneration and is currently the only drug of this class used in clinics [3]. It was observed that, in patients with Alzheimer’s disease, memantine improves cognition, behavior, and daily functioning [4,5]. Nevertheless, according to a large-scale meta-analysis, memantine carries with it the significant risk of side effects, including somnolence, weight gain, confusion, and hypertension [6]. In mice, after long-term administration, it evokes an anxiety-like phenotype, possibly linked to the reduction of astrocytic glutamate uptake [7]. At the same time, there are multiple studies that did not report significant negative effects of memantine [8,9]. Moreover, some authors suggest that memantine could be used as an auxiliary therapy for anxiety disorders [10]. Thus, further investigation of potential side effects of memantine, especially under conditions of chronic administration, is warranted.

Our laboratory has focused on mechanisms of spinocerebellar ataxia type 1 (SCA1). This condition belongs to the family of polyglutamine diseases and is caused by the expansion of the polyglutamine tract in the protein Ataxin-1. In our previous studies, we found that excitotoxic processes, characteristic of a SCA1 mouse model, can be mimicked by chronic optogenetic overactivation of Bergmann glia, which expresses channelrhodopsin-2 [11]. Using this model, we showed that memantine administration significantly attenuated loss of Purkinje cells and preserved normal Bergmann glia morphology [12]. These data indicate that memantine may have therapeutic potential in SCA1. However, we were concerned that the side effects of memantine could alter the behavior, memory, or motor performance of the mice and that these side effects may only become evident after sufficiently long treatment.

Here, we examined motor performance and anxiety-like behavior in mice after chronic (9 weeks) memantine consumption. We demonstrate a significant negative impact of memantine on mice in the rotarod test, a negative effect on memory retention and an anxiety-like phenotype in the elevated cross paradigm. In vitro, after chronic memantine administration, but without the drug being present during recording, we find that the amplitude of field Excitatory Post Synaptic Currents (fEPCS) was greatly reduced in CA1 and CA3 areas of the hippocampus. In addition, chronic memantine increased the decay time of fEPSCs in CA1.

## 2. Materials and Methods

All procedures for the care and treatment of animals were carried out according to the Krasnoyarsk State Medical University and Russian public standard (33215–2014) regulations and were approved by the local ethical committee. Every effort was made to minimize animal suffering and reduce the number of animals used in this study. Twelve-week-old CD-1 IGS WT mice of both sexes (Charles River Laboratories) were used in this study. Animals were kept on a 12 h light/dark cycle with free access to food and water. We used 17 control and 24 memantine treated mice in experiments.

### 2.1. Memantine Administration

Memantine (Sigma-Aldrich, St. Louis, MO, USA, cat. No M9292) was added to drinking water at 0.5 mg/mL, as described previously [12]. On average, mice consumed daily ~5.8 ± 0.2 mL/30 g of mouse body weight (consistent with [13]). The drinking bottle was replaced once a week during the 9-week period, and the intake was confirmed. Thus, the average calculated daily dose of memantine was approximately 90 mg/kg∙day [14]. Memantine consumption did not change the body weight of examined mice (Appendix A), *p* = 0.22, (n = 14 mice in both groups). We did not notice any pathological changes in these animals.

### 2.2. Behavioral Testing

#### 2.2.1. Rotarod Test

Motor coordination was assessed on 5 consecutive days by rotarod tests with an acceleration protocol (3 min acceleration from 0 to 30 revolutions per minute (rpm)), which included 4 trials with a 30 min inter-trial interval. The rod (Rota-Rod Neurobotics, Moscow, Russia) consisted of a gridded plastic rod (3 cm in diameter, 10 cm long) flanked by two large round plates (50 cm in diameter). We recorded the time that the mice spent on the rotarod with a cutoff of 300 s. The time spent on the rod was averaged across all the trials on the same day and that number was then used in the statistical analysis.

#### 2.2.2. Contextual and Cued Fear Conditioning

A fear conditioning protocol was carried out on three consecutive days, according to the standard approach [15]. The test device was an acrylic square chamber (size 33 × 25 × 28 cm), which was placed inside a soundproof chamber (size 170 × 210 × 200 cm) to minimize external noise during testing. A backlight (100 lux LEDs) and a speaker were mounted above the test chamber to supply a conditioned stimulus (CS). The grid floor was connected to a generator (Ugo Basile, Gemonio, Italy) to provide an electrical signal as an unconditioned stimulus (US). On day 1, conditions for freezing were created (conditioning test). Mice were placed into the chamber and allowed to freely explore for 120 s. After that, an auditory CS was presented (white noise, 55 dB) for 30 s, and a 0.3 mA foot shock was delivered as an US during the last 2 s of the white noise. To establish the association, the CS-US pairing was repeated three times per session at 120, 240, and 360 s after the beginning of the trial. The testing process was recorded using the ANY MAZE animal video analysis system (Behavior Tracking Software, Stoelting, Chicago, IL, USA), and the dynamics of acquisition of the association of CS and US were tracked and analyzed for each time interval (0 to 120 s, 120 to 240 s, 240 to 360 s, 360 to 480 s). We measured freezing episodes and time freezing (in s) during the test. The context test was performed 24 h later, whereby mice were placed into the same chamber for 300 s in the absence of any stimuli. A cued test was performed on day 3 in a new environment. This test chamber differed in terms of wall color, floor structure (no grid), and 30 lux illumination, providing a new context from the chamber associated with foot shocks. The test consisted of a 180 s period of mice exploring the new environment, followed by 180 s CS but without the US. The percentage of the duration of the freezing episodes during each time interval was determined as an indicator of fear memory. Freezing is a commonly used behavioral response interpreted as index of fear in mice [15,16].

#### 2.2.3. Elevated Plus Maze

The elevated plus maze (EPM) was used to investigate anxiety-like behavior [17]. The maze consisted of two open arms (6 cm × 32 cm) and two closed arms (6 cm × 32 cm with 19 cm tall opaque walls) with a center area 6 cm × 6 cm and was raised 54 cm above the floor. The surrounding room was dark, and the maze was lit by overhead lights. The mouse was placed in the center of the maze facing an open arm and allowed to explore for 5 min. During this time, the movement was detected automatically using ANY MAZE animal video analysis system. The time spent in closed arms was recorded. The maze was cleaned with 70% ethanol before each trial [18].

#### 2.2.4. Tail Suspension Test

The tail-suspension test (TST) is used to reveal behavioral despair and for assessing effects of antidepressants in mice [19,20]. A mouse was suspended by its tail at more than 10 cm from the floor for 6 min. Periods and time of complete immobility were detected using the ANY MAZE video analysis system. Mice that climbed their tail or fell off the hanger were excluded from analysis.

### 2.3. Electrophysiology

#### 2.3.1. Acute Slice Preparation

Once deep anesthesia was achieved with Zoletil (50 mg/kg i.p.; Virbac, Carros, France), the mouse was decapitated, the brain extracted, and immediately placed into ice-cold Ringer’s solution, perfused by 95% O_2_ + 5% CO_2_, for 1 min. Coronary sections, 350 μm thick, including the hippocampus, were obtained using a Thermo Scientific MicromHM650V vibratome. The slices were cut in Ringer’s solution (mM: 234 sucrose, 26 NaHCO_3_, 2.5 KCl, 1.25, NaH_2_PO_4_, 11 glucose, 10 MgSO_4_, and 0.5 CaCl_2_) at 4 °C with a constant supply of 95% O_2_ + 5% CO_2_ mix [21].

#### 2.3.2. Field Excitatory Postsynaptic Potential (fEPSP) Recording

Brain slices were placed in a chamber mounted on a microscope stage (Olympus BX51WI) and perfused with oxygenated (95% O_2_ + 5% CO_2_) extracellular solution containing (in mM) 125 NaCl, 2.5 KCl, 2 CaCl_2_, 1 MgCl_2_, 1.25 NaH_2_PO_4_, 26 NaHCO_3_, and 10 glucose at pH 7.2 at a rate of ~2 mL/min. To assess the total synaptic activity of neurons, fEPSPs were recorded from CA1 and CA3 zones of ventral hippocampus using 5–10 MΩ borosilicate glass electrodes filled with extracellular solution, as described previously [22]. Neuronal activity was elicited by electrical stimulation (0.1 ms duration, 0.33 Hz) using a 5 MΩ platinum iridium electrode placed on Schäffer collateral. Data registration (3 kHz filter) and conversion into a digital format were carried out using an HEKA EPC10 amplifier. fEPSP amplitude, rise time, decay time, and paired-pulse facilitation (PPF) ratio were measured. The PPF was recorded with 50 ms interval between first and second fEPSPs and measured as a ratio of second fEPSP to first fEPSP.

### 2.4. Statistical Analysis

Pooled data are expressed as mean values ± S.E.M. with a 95% confidence interval. We used basic statistical functions of free open-source software environment R to perform the statistical analyses. The differences between the individual groups were estimated by the ANOVA model and the Tukey-Kramer test, which is applicable for *p*-values adjustment if the samples have unequal size. Differences were considered significant at *p* < 0.05.

## 3. Results

### 3.1. Chronic Administration of Memantine Affects Rotarod Performance

As mentioned in the introduction, we are considering the potential of memantine for treatment of neurodegeneration caused by SCA1 and similar diseases that lead to motor disturbances among other symptoms. To test whether long-term treatment with memantine could affect motor performance on its own, here, we treated mice with memantine for 9 weeks and assessed their rotarod performance (Figure 1A). Both treated and untreated mice improved their performance from day 1 to day 2 of testing, and untreated mice then gradually improved their performance until day 4, when it stabilized and remained approximately the same on day 5. However, memantine-treated mice were unable to improve their rotarod retention time above their scores on day 2, and on days 4 and 5 performed significantly worse than control mice. For example, on day 5, control mice stayed on the rod from 233.7 ± 11.3 s (16 animals), while the memantine treated mice for only 176.0 ± 24.1 s (11 animals; *p* = 0.024; Figure 1C).

### 3.2. Long-Term Memantine Administration Affects Contextual Memory Retrieval

We used contextual fear conditioning to investigate possible effects of chronic administration of memantine on memory. Naïve and memantine treated mice were trained in a cued conditioning protocol for 3 days (Figure 2A). There was no statistically significant difference in the number of freezing episodes between the two groups on any of the days; *p* = 0.16 (Figure 2B). However, memantine-treated mice demonstrated significantly shorter freezing times on day 3 when retention of cues memory was assessed (205.8 ± 12.6 s, 24 mice, vs. 248.7 ± 9.7 s, 17 mice; *p* = 0.0099) (Figure 2C). Thus, chronic memantine consumption disrupted cued memory retrieval.

### 3.3. Long-Term Memantine Administration Prevents Adaptation to Anxiogenic Environment

Upon careful analysis of the rotarod data, we concluded that suppression of the new motor skill acquisition was not necessarily the main reason for the poor performance of memantine-treated mice. During the test, mice often jump and try to escape from the rod; however, this behavior in naïve mice typically ceases after the second day when mice become calmer and stay on the rod for the duration of the test. However, memantine-treated mice exhibited this type of anxious behavior until the last day of the test. For this reason, we tested their anxiety using an elevated plus maze protocol [17]. Mice treated with memantine spent the same time in closed arms as their naïve counterparts on the first and second days of training (Figure 3A). However, from the third day on, memantine-treated mice spent significantly more time in the closed arms, which is an accepted index of anxiety (for example on day 3, 163.9 ± 20.2 s vs. 79.6 ± 16.3 s, *p* = 0.008). The difference was seen until day 5 (133.4 ± 21.0 s and 61.0 ± 21.9 s, respectively, *p* = 0.036; Figure 3A,B). Interestingly, these differences followed the dynamics seen in rotarod performance (Figure 1C).

### 3.4. Long-Term Memantine Administration Induces Signs of Behavioral Despair in TST

The previous results suggested that chronic memantine could enhance anxiety under some conditions or make mice prone to depression-like behavior. We therefore performed TST as an index of behavioral despair frequently used in studies of antidepressants [20,23]. Animals treated with memantine showed significantly more immobility episodes (14.9 ± 2.0, 13 animals vs. 8.9 ± 1.4, 14 animals, respectively, *p* = 0.026; Figure 4A). Moreover, immobility periods in these animals were significantly longer (102.1 ± 14.5 and 43.4 ± 6.2, respectively, *p* = 0.0016; Figure 4B).

Taken together, these results indicate that chronic memantine administration hinders adaptation to mildly stressful conditions, such as free movement on a x-maze track. TST suggests that it induces signs of behavioral despair, which is usually interpreted as and an element of a depressive phenotype [20].

### 3.5. Memantine Disrupts Synaptic Transmission in Ventral Hippocampus

It was demonstrated previously that chronic antagonism of NMDA receptors evoke anxiety in mice [24]. Anxiety is a complex phenomenon involving multiple structures of the brain in various manifestations of anxiety, the hippocampus being one of them. Here, we recorded fEPSPs from ventral hippocampus, which has been implicated in anxiety behavior by previous studies [25,26,27]. Importantly, the hippocampi were taking from mice that had been chronically treated with memantine, but, during these experiments, memantine was absent from the perfusion media. Thus, the effects seen should be attributable to lasting modifications of either glutamatergic synapses or astrocytic glutamate removal systems. When single stimuli were applied in CA1 and CA3 areas of ventral hippocampus, memantine pre-treatment significantly reduced fEPSP amplitudes. In CA1, fEPSPs were reduced from 0.38 ± 0.02 mV to 0.29 ± 0.02 mV (*p* = 0.005) and in CA3 from 0.36 ± 0.04 mV to 0.22 ± 0.03 mV (*p* = 0.028; Figure 5A, B left panels; Table 1 and Table 2). fEPSP decay time after application of individual stimuli did not change significantly (Table 1 and Table 2). To load the synapse with glutamate and better reveal the workings of the uptake system, we facilitated neurotransmitter release from presynaptic terminals using trains of 10 electrical stimuli (frequency 10 KHz). Under such conditions, we found prolongation of the decay phase half time from 4.2 ± 0.3 ms to 5.9 ± 0.5 ms (*p* = 0.006) in area CA1 (Figure 5A and Table 1). In CA3, we did not find this effect (*p* = 0.58; Figure 5B and Table 2).

In vivo, memantine remains in plasma for up to 132 h after oral administration [28] and can be detected in mouse brain 100 min after administration [29]. According to microdialysis data, the half-life of memantine in rodent brain is approximately 3 h [30]. Such time course implies that memantine was present in the tissues during our in vivo mouse training sessions. However, memantine is effectively washed from the tissue during slice preparation and perfusion in vitro [31]. Therefore, the effects that we observed in fEPSP recordings most likely cannot be ascribed to direct NMDA receptor inhibition. Rather, these data strongly suggest that memantine causes lasting alterations in glutamatergic synaptic transmission in the ventral hippocampus.

## 4. Discussion

Memantine is approved by the FDA for treatment of Alzheimer’s disease but potentially may be useful for the treatment of other neurodegenerative conditions, as well. We previously reported that memantine reduces the signs of Purkinje Cell and Bergmann Glia damage in our model of cerebellar neurodegeneration induced by optogenetic overactivation of Bergmann Glia [11]. We therefore suggested that memantine should be tested for treatment of conditions such as spinocerebellar ataxias. However, this required to first establish that memantine as such did not disrupt motor behavior. Here, we examined rotarod performance of mice after 9 weeks of memantine consumption (Figure 1). Mice were tested for 5 successive days. Surprisingly, we found that memantine-treated mice performed at similar levels to controls on days 1 and 2, but from day 3 their performance did not improve, in contrast to control animals. Such a prominent negative effect may cause failure at clinical trials and requires further investigation. The most common explanation, disruption of cerebellar function, seemed unlikely, because on days 1 and 2 memantine-treated mice did not differ from controls.

An alternative possibility is that memantine disrupted acquisition of the skills required to stay on the rod. This possibility is supported by two lines of evidence. First, we saw a deficit in cued memory retrieval where memantine-treated mice remembered the auditory cue worse than controls (Figure 2), as demonstrated by shorter freezing times. Second, we found evidence of reduced glutamate transmission in the CA1 and CA3 area of the hippocampus. While such a reduction of fEPSP is not a direct evidence of memory disruption, it can be argued that reduction in glutamatergic transmission in the CA1 area may be expected to negatively impact on hippocampal memory-related functions. Interestingly, when we used trains of 10 stimuli (Figure 5B), we detected a significant prolongation of the decay phase of the fEPSP in the CA1 area. The most logic explanation of this effect is reduction of the glutamate clearance from the extracellular space, which points to the involvement of astrocytes, the cells which play a key role in glutamate clearance.

However, there was also another possible explanation for the reduction of the time on rotarod in memantine-treated mice. During the rotarod training, these mice tended to stand on their hind paws, trying to escape. Such behavior was also seen in control mice up to day 2 but then ceased, while it was observed throughout the whole training period in memantine-treated mice. We considered this as a sign of persistent anxiety or perhaps a lack of adaptation to this anxiogenic environment and used the elevated plus maze to further investigate anxiety-like behavior. We found a pattern of persistent anxiety behavior, similar to that in the rotarod protocol. From day 3 onwards, control mice spent less time in closed arms than the memantine-treated mice, which spent approximately the same time in closed arms on all days of training (Figure 3). This can be seen as an adaptation of control mice to these mildly stressful conditions when mice learn that open arms do not represent danger. Different behavior of the memantine-treated group points to the remaining anxiety in these animals on days 3, 4, and 5, which may be a consequence of the loss of adaptation to the apparatus.

We then used the acute and more stressful tail suspension test to investigate signs of behavioral despair. In the memantine-treated group, mice had more immobility episodes and longer immobility periods (Figure 4). Such changes are commonly seen as indications of depression-like behavior and have been commonly used in the past for looking at the effects of antidepressants [20,32].

In summary, our data supports the idea that long-term consumption of memantine can reduce the adaptation to anxiogenic environments, likely related to the impact on memory retention discussed above. We believe that such an effect in humans could be a serious handicap because we often experience fear of novel environments but then learn that it is safe.

To the best of our knowledge, no previous study investigated possible side effects of memantine after prolonged administration (9 weeks), although in clinics this drug has been used for many years. Interestingly, single doses of memantine were reported to produce “antidepressant-like” effects in TST (2.5–15 mg/kg). Acute doses up to 30 mg/kg affected rearing, ambulation, and grooming. Under some conditions, ataxia and stereotypy were detected with 30 mg/kg [33].

Our results indicate that when memantine is administered for months, it induces changes that are best described as compromised ability to cope with stress. This may reflect the inability to remember that the experimental environment does not pose a threat and may be a consequence of a learning deficit. We also reveal an element of behavioral despair, often associated with depression phenotype in TST. These effects could be linked to persistent changes in synaptic transmission in the hippocampus and, possibly, other areas of the CNS. We believe that these effects need to be taken into account when considering memantine as a putative new medication in neurodegenerative diseases.

## Figures and Tables

**Figure 1 brainsci-12-00495-f001:**
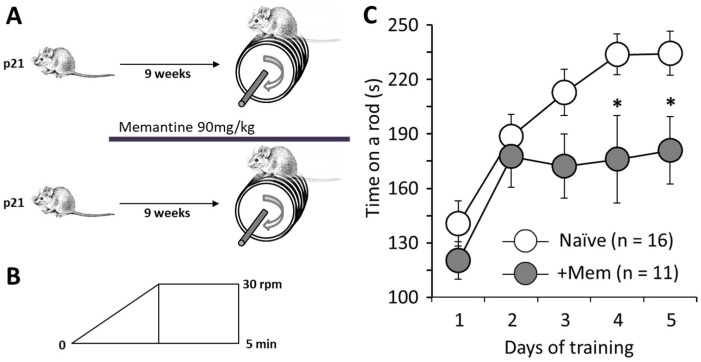
Long-term administration of memantine impairs the motor behavior in mice. (**A**) Experimental groups: control mice did not receive any medication, while the test group consumed memantine (up to approximately 90 mg/kg∙day in drinking water) from postnatal day 21 (p21) for 9 weeks. (**B**) Accelerated rotarod protocol. Speed increased from 0 to 30 rpm during first 3 min and then kept stable for 2 min. (**C**) Averaged data of rotarod performance in control mice (opened circles) and mice treated with memantine (grey circles) over 5 days of testing. (* *p* < 0.05; *t*-test, followed by post-hoc Tukey’s HSD test for *p*-values adjusted for multiple comparisons; n is number of mice in each group).

**Figure 2 brainsci-12-00495-f002:**
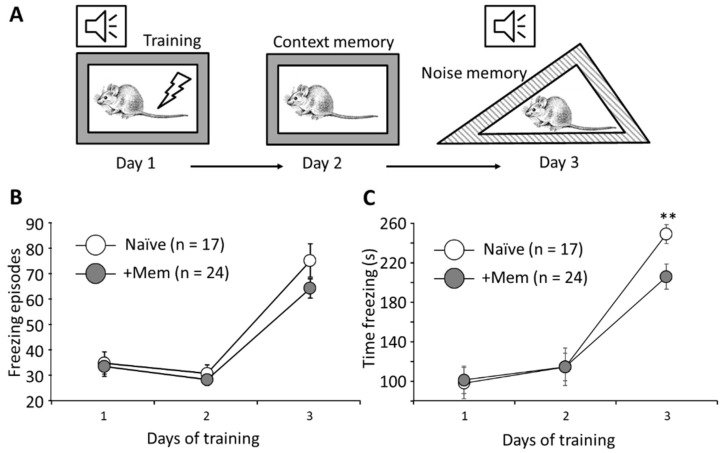
Memantine impairs memory in a fear conditioning paradigm. (**A**) Schematic of the fear conditioning procedure. Day 1: Conditioning consisted of three electric shocks after white noise presentation. Day 2: Context memory test. Day 3: Cued memory test. (**B**) Number of freezing episodes during and after training. The number of freezing episodes on day 3 after training was not significantly changed by memantine (*p* > 0.05). (**C**) Total time of freezing episodes induced by the auditory challenge on day 3 was significantly reduced in memantine-treated mice (** *p* < 0.001; *t*-test, followed by post-hoc Tukey’s HSD test for *p*-values adjustment for multiple comparisons; n is number of mice in each group).

**Figure 3 brainsci-12-00495-f003:**
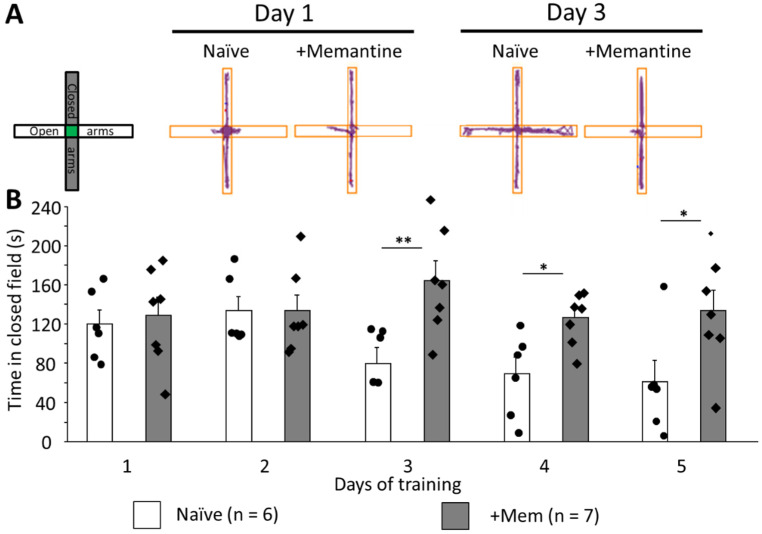
The elevated plus maze test reveals persistent anxiety-like behavior in mice after chronic memantine administration. (**A**) Cumulative movement maps of mice in the arms of the elevated plus maze at day 1 (left panel) and 3 (right panel) of training. (**B**) Time spent in the closed arms. Individual data points are shown as closed circles for control mice and closed diamonds for mice after chronic memantine consumption. By day 3, control mice overcame their anxiety and started exploring open arms while memantine-treated mice remained largely in the closed arms (* *p* < 0.05; ** *p* < 0.001; n is number of mice in each group).

**Figure 4 brainsci-12-00495-f004:**
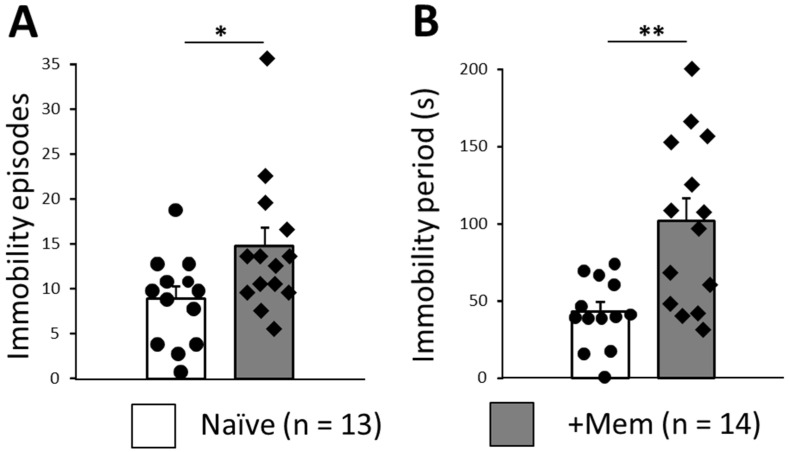
Tail suspension test demonstrates increased anxiety-like behavior in mice chronically treated with memantine. (**A**) Immobility episodes and (**B**) immobility period during 360 s total test duration. Individual data points are shown as closed circles for control mice and closed diamonds for memantine treated mice (* *p* < 0.05; ** *p* < 0.01).

**Figure 5 brainsci-12-00495-f005:**
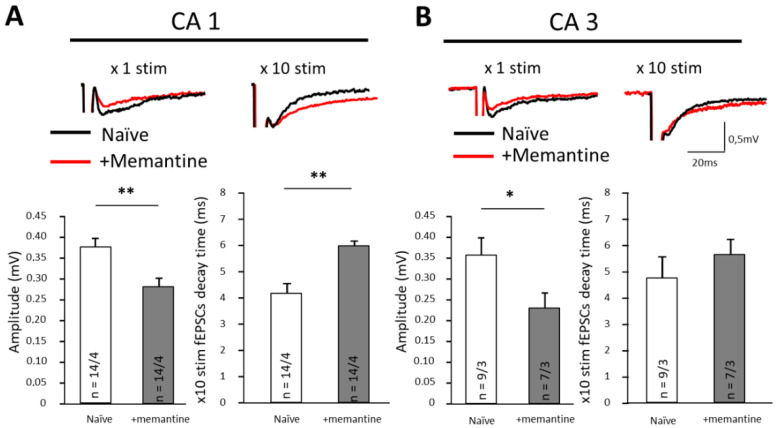
Chronic memantine pre-treatment affects synaptic transmission in hippocampus. (**A**) Recordings from the CA1 area. (**B**) Recordings from the CA3 area. Upper panels—representative traces of fEPSPs after one (×1) stimulus or ten (×10) stimuli in slices from control mice (black traces) and slices from mice after chronic memantine consumption (red traces). Lower panels—average amplitudes of fEPSPs after ×1 stimulus (left) and average decay times of fEPSPs after ×10 stimuli (right). fEPSPs evoked by a single synaptic stimulus were significantly reduced in both areas of the hippocampus (* *p* < 0.05; ** *p* < 0.01). Note that at the time of recording memantine was not present in the bath and would have been largely washed out from the tissue in the course of slice preparation and preincubation.

**Table 1 brainsci-12-00495-t001:** Amplitudes and kinetics of fEPSPs in the CA1 area of hippocampus. Are shown the decay time after one (×1 stim.) or ten (×10 stim.) stimuli. Asterisks indicate statistically significant differences (** *p* < 0.01; n are numbers of tested areas and animals (areas/animals)).

CA1	Amplitude (mV)	Rise Time (ms)	Decay Time (×1 stim.)	Decay Time (×10 stim.)
Control (n = 14/4)	0.38 ± 0.02	2.8 ± 0.4	7.9 ± 0.5	4.2 ± 0.3
+Memantine (n = 14/4)	0.29 ± 0.02 **	2.2 ± 0.2	7.8 ± 0.7	5.9 ± 0.5 **

**Table 2 brainsci-12-00495-t002:** Amplitudes and kinetics of fEPSPs of the ventral hippocampal CA3 area. Are shown the decay time after one (×1 stim.) or ten (×10 stim.) stimuli. Asterisks indicate statistically significant differences (* *p* < 0.05; n are numbers of tested areas and animals (areas/animals)).

CA3	Amplitude (mV)	Rise Time (ms)	Decay Time (×1 stim.)	Decay Time (×10 stim.)
Control (n = 9/3)	0.36 ± 0.04	2.1 ± 0.1	5.2 ± 0.6	4.9 ± 0.9
+Memantine (n = 7/3)	0.22 ± 0.03 *	1.7 ± 0.2	4.9 ± 1.1	5.4 ± 0.5

## Data Availability

Not applicable here.

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
