# Peer review of "Memantine Disrupts Motor Coordination through Anxiety-like Behavior in CD1 Mice"

_brainsci, 2022, doi:10.3390/brainsci12040495_

Round 1
Reviewer 1 Report
This study provides evidence based on behavioral experiments in mice to discuss side effects that should be noted when considering memantine as a presumptive drug for neurodegenerative diseases. The experimental method is reasonable and the consideration of the results is also appropriate. This data should be widely considered and is scientifically important.
Major Comments
Methods
Please provide the total number of mice used to generate the data in this manuscript. In addition, the gender of the mouse should be stated. Did the mice change their body weight after 9 weeks of memantine administration? How is the health of the mice after administration? These things should be mentioned in the section of Material and Methods.
Author Response
Thank you very much for comments! They improove the manuscript. We answred to your comments
- Q: Please provide the total number of mice used to generate the data in this manuscript.
A: We corrected the explanation in the Methods section.
- Q: In addition, the gender of the mouse should be stated.
A: We corrected the explanation in the Methods section.
- Q: Did the mice change their body weight after 9 weeks of memantine administration?
- A: We added this information in Sup. Fig. 1 (see methods).
- Q: How is the health of the mice after administration? These things should be mentioned in the section of Material and Methods.
A: We corrected the explanation in the Methods section.

Reviewer 2 Report
The manuscript entitled “Memantine Disrupts Motor Coordination through Anxiety Like behavior in CD1 Mice” by Shuvaev and coworkers is focusing a very important and actual topic related to therapeutic improvements introducing the memantine in the treatment of CNS disorders. The basic idea is original and directed towards the side effects of memantine administration. Some concerns should be carefully addressed (see below) before the consideration of overall merit.
In the Abstract the authors should exclude what they have done in previous research and focus on actual experimental design.
The number of animals should be mentioned in the MM section. Please explain why the number of animals is different for each test?
The statistical analysis seems to be inappropriate by means of unequal sample sizes (Tukey post hoc is inapplicable), and ANOVA should be applied.
The authors conducted repeated measurements in EPM (for five days in a row), the animals learn and habituate to the apparatus, and lose motivation to explore the already “seen” maze, which also would result in lost sensitivity of key outcome measures (this could interfere with cognitive decline and even result in the opposite conclusions). So, the intersession intervals should be much longer.
The authors stated that TST is a method for assessing depression and anxiety, the mentioned reference doesn’t support that statement. Also, the authors should comment the results from TST in a manner according to the test’s purpose, since this test is not a good choice for anxiety estimation.
Author Response
We are really appreciate your comments! They helped us improve the manuscript. We answered to your comments.
-
Q: In the Abstract the authors should exclude what they have done in previous research and focus on actual experimental design.
A: We have rewritten and shortened the abstract. For context, we left just one sentence to explain the importance of memantine administration in case of cerebellar neurodegeneration. However, we would like to stress that as such, the reported action of memantine will have implications for its introduction as therapy for any disease, not only SCA1.
- Q: The number of animals should be mentioned in the MM section. A: We added this information in the Methods section.
- Q: Please explain why the number of animals is different for each test?
A: We wanted to minimize numbers of animals used while achieving statistical power. Different tests will have different variance and effect sizes.
- Q: The statistical analysis seems to be inappropriate by means of unequal sample sizes (Tukey post hoc is inapplicable), and ANOVA should be applied.
A: We used the R software that applies the Tukey-Kramer test (which is appropriate for unequal sizes of samples) on the ANOVA model by TukeyHSD command. We corrected the explanation in the Methods section.
- Q: The authors conducted repeated measurements in EPM (for five days in a row), the animals learn and habituate to the apparatus, and lose motivation to explore the already “seen” maze, which also would result in lost sensitivity of key outcome measures (this could interfere with cognitive decline and even result in the opposite conclusions). So, the intersession intervals should be much longer.
A: Thank you for this comment. We appreciate that repetitive EPM tests lead to loss of motivation to explore the already “familiar” maze. We used this test not in in order to study the curiosity or spatial cognitive function in mice, but to assess the anxiety-like phenotype. This test is a standard approach for this condition since 1985 (Pellow et al., 1985). We used the EPM test for 5 days in a row to connect anxiety with altered motor behavior in memantine treated mice. Since testing of motor behavior was carried out on 5 consecutive days (with a prominent effect at days 3-5), intersession intervals could not be longer. We agree with the reviewer that loss of motivation to explore the plus maze was evident in all tested groups. For this reason, the continued preference for dark arms in memantine-treated mice specifically reflects a difference in a mouse’s ability to adapt to a stressful situation.
- Q: The authors stated that TST is a method for assessing depression and anxiety, the mentioned reference doesn’t support that statement. Also, the authors should comment the results from TST in a manner according to the test’s purpose, since this test is not a good choice for anxiety estimation.
A: Anxiety and depression are thought to be relatively strongly correlated comorbidities in both humans and animals. For this reason behavior tests for these conditions are the same. TST is extremely common test for depression and anxiety for decades (Kennedy et al., 2008). Recently, a study mentioned that anxiety levels are substantially affected even in handled mice, when moved from cage to cage or to experimental box (Ueno et al., 2020). For this reason, TST is a straightforward and robust test to examine anxiety-like behavior in mice. We corrected the explanation in the Methods section.
Reference:
- Ueno, H., Takahashi, Y., Suemitsu, S. et al. Effects of repetitive gentle handling of male C57BL/6NCrl mice on comparative behavioural test results. Sci Rep 10, 3509 (2020). https://doi.org/10.1038/s41598-020-60530-4
- Kennedy, S. H. Core symptoms of major depressive disorder: relevance to diagnosis and treatment. Dialogues Clin. Neurosci. 10, 271–277 (2008).
- Pellow S, Chopin P, File SE, Briley M. Validation of open: closed arm entries in an elevated plus-maze as a measure of anxiety in the rat. J Neurosci Methods. 1985;14:149–167.

Round 2
Reviewer 2 Report
Although the authors responded to some specific comments in an acceptable manner, still there are some substantial concerns that were not addressed appropriately:
The tail-suspension test (TST) is a standard method for assessing depression and anxiety phenotypes in mice [20].
I should state again that TST is not an appropriate method for assessing anxiety, and I strongly confirm that the mentioned reference doesn’t support that statement (the anxiety is not mentioned in the article).
In response to my comments, instead of required intervention, the authors stated that “TST is extremely common test for depression and anxiety for decades (Kennedy et al., 2008).”. This statement does not stand and is not supported by literature data. Even more, the offered reference is focusing on the results of clinical trials, with no data considering animal experimental models, behavioral testing, and especially not “common test for depression and anxiety for decades”.
The patterns of anxiety-like behavior observed in EPM test appears to be the result of confronting the curiosity and inherited explorative reaction of rodents with the fear of the novelty environment. Therefore, any protocol that could affect the exploration mood, such as familiarization, can principally mimic the anxiogenic response (“the elevated plus-maze test requires inexperienced subjects to obtain reliable results” [https://doi.org/10.1016/S0166-4328(97)02286-9]). Therefore, I can confirm that this experimental design is not appropriate for such an investigation.
Author Response
We are grateful to the reviewer for their effort to review our manuscript and hope that we have now addressed the criticisms.
The reviewer raised again concerns about TST.
Upon reflection we agree that it was not accurate to call TST a method for assessing depression and anxiety. Thank you for drawing our attention to this issue. This statement rather reflected the fact that TST has been used by numerous studies (hundreds of them) which looked at the effects of antidepressants but indeed, this may be seen as not an accurate definition. We have now corrected the text and everywhere where TST is mentioned refer to it as a procedure which assesses behavioural despair. This definition is in line with the most reputed sources, such as papers by Prof E Nestler (we introduced new citations). We hope that the referee will agree that this is an accurate definition and would like to stress that TST, simple as at is, is still used widely around the world. Fundamentally, it does not affect our conclusions pointing at potential, previously unreported negative effects of memantine after chronic administration.
Regarding the criticisms of the EPM test, we believe that the reviewer could have misinterpreted our thoughts. We tried to clarify our point in the revised version of the manuscript which now reads:
During the rotarod training, these mice tended to stand on their hind paws, trying to escape. Such behavior was seen up to day 2 in control mice also but remained throughout the whole training period in memantine-treated mice. We considered this as a sign of persistent anxiety, or, perhaps the lack of adaptation to this anxiogenic environment. The pattern of persistent anxiety behavior was similar to that in the rotarod protocol. From day 3 onwards, control mice spent less time in closed arms than the memantine-treated mice which spent approximately the same time in closed arms on all days of training (Fig. 3). This can be seen as adaptation to these mildly stressful conditions when mice learn that open arms do not represent danger. Different behaviour of memantine treated group points to the remaining anxiety in these animals days 3, 4 and 5 which may be a consequence of the loss of adaptation to the labyrinth.
We then used the acute and more stressful tail suspension test to investigate signs of behavioural despair. In the memantine-treated group, mice had more immobility episodes and longer immobility periods (Fig. 4). Such changes are commonly seen as indications of depression-like behaviour and have been commonly used in the past for looking at the effects of antidepressants.
We thank the reviewer for helping us to improve the clarity and would like to stress here that such an effect could have negative effects on human every day behaviour because only by experience we learn what is dangerous and what is not.
With regards to the reviewer’s comments about repeated application of the EPM and cited paper by Espejo (1997), we respectfully disagree, that this test cannot be used more than once. This depends on the purpose and what the Espejo paper shows is that indeed, upon repeated placing into the labyrinth the behaviour of mice changes but the conclusion they made only means that one cannot compare the effects in mice who have never seen the labyrinth with mice who have been there several times. Our experiments show exactly the same (although, for reasons we cannot explain in Espejo paper mice seemed to stay more in the closed arms after 3 days of training). However, we do not compare untrained mice with those who have been placed in the EPM several times, we have two groups who follow exactly same trajectory. What we see is that control mice after 3 days probably realise that the open arms are not dangerous and spend more time there while after memantine they remain mainly in closed arms as if they were still anxious and unable to adapt.
In the paper we write:
Our results indicate that when memantine is administered for months, it induces changes which are best described as compromised ability to cope with stress.
We believe that such an effect in humans could be a serious handicap because we often experience fear of novel environments but then learn that it is safe. One can see that as a learning deficit, but in this specific case it is difficult to the line between a learning deficit and state of anxiety.
We hope that the corrections we have made will be seen as sufficient.
